# Novel Hybrid Inulin–Soy Protein Nanoparticles Simultaneously Loaded with (-)-Epicatechin and Quercetin and Their In Vitro Evaluation

**DOI:** 10.3390/nano13101615

**Published:** 2023-05-11

**Authors:** Jocelyn C. Ayala-Fuentes, Maryam Soleimani, Jonathan Javier Magaña, Jose Mario Gonzalez-Meljem, Rocio Alejandra Chavez-Santoscoy

**Affiliations:** 1Tecnologico de Monterrey, Escuela de Ingenieria y Ciencias, Campus Monterrey, Ave. Eugenio Garza Sada 2501 Sur, Monterrey 64849, Mexico; 2Tecnologico de Monterrey, Escuela de Ingenieria y Ciencias, Mexico City 14380, Mexico; 3Laboratorio de Medicina Genómica, Instituto Nacional de Rehabilitación-Luis Guillermo Ibarra Ibarra (INR-LGII), Mexico City 14389, Mexico

**Keywords:** nanoparticle, bioavailability, quercetin, epicatechin, inulin, soy protein, spray drying

## Abstract

(-)-Epicatechin and quercetin have attracted considerable attention for their potential therapeutic application in non-communicable chronic diseases. A novel hybrid inulin–soy protein nanoparticle formulation was simultaneously loaded with (-)-epicatechin and quercetin (NEQs) to improve the bioavailability of these flavonoids in the human body, and NEQs were synthesized by spray drying. After process optimization, the physicochemical and functional properties of NEQs were characterized including in vitro release, in vitro gastrointestinal digestion, and cell viability assays. Results showed that NEQs are an average size of 280.17 ± 13.42 nm and have a zeta potential of −18.267 ± 0.83 mV in the organic phase. Encapsulation efficiency of (-)-epicatechin and quercetin reached 97.04 ± 0.01 and 92.05 ± 1.95%, respectively. A 3.5% soy protein content conferred controlled release characteristics to the delivery system. Furthermore, NEQs presented inhibitory effects in Caco-2, but not in HepG-2 and HDFa cell lines. These results contribute to the design and fabrication of inulin–soy protein nanoparticles for improving the bioavailability of multiple bioactive compounds with beneficial properties.

## 1. Introduction

Chronic non-communicable diseases (NCDs) are conditions that result in long-term health consequences and long-term treatment and care, including cancers, cardiovascular disease, diabetes, and chronic lung illnesses [1]. In the last decades, the bioactive properties of two members of the flavonoid family of natural compounds, (-)-epicatechin and quercetin, have garnered considerable attention and prompted re-search into their application as potential preventive therapeutics against NCDs [2].

(-)-Epicatechin is found in green tea, black tea, grapes, berries, nuts, beans, some cereals (barley and sorghum), and spices such as curry, cinnamon, and cacao. It has diverse biological properties such as anti-microbial, anti-inflammatory, antitumor, cardioprotective and anti-diabetic activities [3]. However, recent findings suggest that (-)-epicatechin has poor in vivo bioavailability and poor gastrointestinal tract (GIT) absorption due to its partial degradation in the stomach [3]. Likewise, it has been reported that low levels of (-)-epicatechin are absorbed and found in the circulation as glucuronides, methylated, and sulfated forms [4].

Quercetin (3,3,4,5,7-pentahydroxyflavone) is one of the most abundant dietary flavonoids [5]. It has strong antioxidant activity, as well as anti-cancer, anti-inflammatory, anti-diabetic, cardioprotective, antibacterial, and hepatoprotective properties [6]. However, the poor solubility of quercetin and its crystalline form at body temperatures limit its bioaccessibility and bioavailability, increase its chemical instability, and shorten its biological half-life, all of which may reduce its efficacy when used in the food and pharmaceutical fields [7].

Our research group previously showed that the combination of (-)-epicatechin and quercetin had an enhanced effect on the cytoplasmic redox environment in vitro compared to each flavonoid alone. Moreover, a placebo-controlled study (n = 156) showed that dietary intervention with bread enriched with a 1:1 mixture of (-)-epicatechin and quercetin decreased the number of nuclear abnormalities in patients’ buccal epithelial cells, while also improving several biochemical parameters associated with metabolic syndrome, such as decreased total cholesterol, LDL-cholesterol, total triglycerides, and fasting plasma glucose after three months of daily consumption [8]. Another study conducted in the model organism Drosophila melanogaster showed that short-term treatment with quercetin and (-)-epicatechin resulted in beneficial effects and stimulated resistance to acute γ-irradiation by restoring mitochondrial membrane potential, decreasing reactive oxygen species production, and increasing MAPK activity [9].

There is evidence that encapsulating phenolics and antioxidants in nanoparticles increases their bioavailability by increasing passive absorption from the lumen of the intestine into the lymphatic and blood circulatory system [10]. According to Peñalva et al., [11], the combination of casein and 2-hydroxypropyl-β-cyclodextrin produces nanoparticles that increase the bioavailability of quercetin nine times more than an oral solution of the flavonoid. As for (−)-epicatechin, it has been reported that (−)-epicatechin-loaded lecithin-chitosan nanoparticles enhanced the cytotoxic effect of (−)-epicatechin against breast cancer cell lines [12]. Additionally, it was reported that (+)-catechin and quercetin dual-loaded nanoparticles outperformed native pharmaceuticals (single or mixed) and single drug-loaded nanoparticles in therapeutic effectiveness [13].

Among the various nanoparticle formulation strategies, protein-based nanoparticles have received considerable attention due to their biodegradability and tunable properties. Protein polymers are natural macromolecules and thus biocompatible, which can be obtained from renewable sources through well-known extraction processes and are widely available commercially [14]. Protein-based nanoparticles have been used as a strategy to improve the solubility and stability of poorly soluble bioactive compounds, thus facilitating their use in oral drug delivery applications [15]. One example of protein-based nanoparticles is soy protein, which is composed of globular proteins isolated from soybeans, and is one of the most abundant types of plant proteins. Importantly, soy protein has great potential to act as a carrier for poorly soluble bioactive compounds, thanks to its surface hydrophobic nature [16], and as a coating in conjunction with other materials either for protection or for physical or chemical surface modification [14]. Hybrid nanoparticles have advantages compared to nanoparticles made of a single material. In addition to their main component, protein-based nanoparticles can be modified with additional molecules that enhance their properties. For example, different combinations of polysaccharides with proteins, lipids, or synthetic biopolymers can improve bioavailability and release properties.

Inulin is a cheap and natural hydrophilic polysaccharide that possesses non-toxic, biodegradable, and dietary fibrous properties and can be harvested from different plants, vegetables, and fruits [17]. Its structure consists of 2–60 units of linear fructose chains that are highly flexible, thus being an interesting candidate for drug delivery strategies. Importantly, inulin is stable in the gastrointestinal tract (GIT) until it reaches the colon [18] and has been shown to promote gastrointestinal health through the maintenance of the colon’s metabolism and microbiota [19].

The present work aimed to nanoencapsulate (-)-epicatechin and quercetin into a single inulin–soy protein nanoparticulate system to enhance their bioavailability within the GIT. A thorough characterization of the physicochemical properties of the synthesized nanoparticles was carried out, including particle size, surface morphology, zeta potential, and drug loading capacity. Their potential efficacy for NCD treatment through in vitro cell viability assays on cancer and normal cell lines was also evaluated, as well as in vitro release and gastrointestinal digestion experiments.

## 2. Materials and Methods

### 2.1. Materials and Chemicals

Organic inulin from agave powder was purchased from Natura Bio Foods^®^ (≤90% purity), isolated soy protein from NATSA Superfood^®^ (≤90% purity), commercial quercetin (aSquared Nutrition^®^) (≤95% purity), commercial (-)-epicatechin (CocoaVia^®^) (≤17% of purity), quercetin 95% (HPLC, Sigma Aldrich, San Luis, MO, USA), (-)-epicatechin 90% (HPLC, Sigma Aldrich, San Luis, MO, USA), absolute ethanol 99.5% (J.T. Baker), Milli Q (Milli-Q Integral 15 Equipment, Merck México, Estado de Mexico, Mexico), distilled water (Milli-Q Integral 15 Equipment, Merck México, Estado de Mexico, Mexico), acetonitrile 99.95% (DEQ), methanol 99.96% (J.T. Baker), acetone 99.95% (J.T. Baker), HCl acid 37.40% (DEQ), NaOH 99.95% (DEQ), and glacial acetic acid 99.96% (DEQ). The in vitro digestion enzyme was α-amylase A-3176 (Sigma-Aldrich, Monterrey, Mexico), porcine pepsin P-7000 (Sigma-Aldrich, Monterrey, Mexico) and pancreatin P-7545 was added (Sigma-Aldrich, Monterrey, Mexico). For the HPLC protocol the chemicals were Acetonitrile HPLC (VWR Chemicals), water HPLC (TEDIA), formic acid solution 49–51% (grade HPLC, Fluka Analytical). The other chemicals used in this study were of analytical grade and purchased from local suppliers.

### 2.2. Optimization of the Spray Drying Process

To optimize the nanoparticles loaded and not loaded with (-)-epicatechin and quercetin, an experimental design Box–Behnken was carried out. The levels and variables are presented in Table 1, where three independent variables were evaluated: inulin concentrations (*w*/*v*), soy protein concentration dependent of inulin concentration (*w*/*w*), and flavonoid proportion according to inulin concentration. The relations between (-)-epicatechin and quercetin were 1:1 in all experiments, with a total of 17 experiments. The response variables are size (nm), polydispersity index (PDI), and encapsulation efficiency (EE%). The response surface methodology (RSM) was used to minimize the dependent variables.

The recovery performance of the spray drying was calculated considering the next equation:(1)$$\mathrm{Equipment}\;\mathrm{efficient}\;(\%)=\frac{\mathrm{Final}\;\mathrm{mass}\;\mathrm{recovery}\;}{\mathrm{Total}\;\mathrm{solid}\;\mathrm{in}\;\mathrm{initial}\;\mathrm{solution}}\;\times \;100$$

### 2.3. Synthesis of Inulin–Soy Protein Nanoparticles

After optimizing the formulation, as described in Section 2.2, the following 3 nanoparticles were prepared: Inulin–soy protein nanoparticles loaded with (-)-epicatechin and quercetin (hereafter referred to as NEQs); then, the solution was prepared with inulin concentration at 4.5% (*w*/*v*, v = final volume of solution), soy protein at 3% (w/wt, wt = weight of total inulin in the solution), a stock solution of (-)-epicatechin and quercetin (flavonoids proportion 1:1 (-)-epicatechin:quercetin, and then 1:50 proportion of solution:flavonoids). Flavonoid solutions were prepared by weighing 112.5 mg into 25 mL, and then they were titrated to obtain an ethanol solution in water (1:5 *v*/*v*) each, and subjected to sonication (Branson 2510 Ultrasonic Cleaner) at 40 KHz frequency and no more than 25 °C for 20 min. This solution was added dropwise to the inulin and soy protein solution previously dissolved in a volume of 225 mL. It was left undergoing magnetic stirring at a temperature of 40 °C for 30 min, and then the final solution was sonicated for 15 min at room temperature (25 °C) [20]. Blank nanoparticles (hereafter referred to as NBs) were used as a negative control; they consisted of inulin–soy protein without (-)-epicatechin and quercetin. Simple nanoparticles (hereafter referred to as NPs) consisted of the same formulation as NEQs, but without soy protein. For both NBs and NPs, the synthesis methodology was the same as for NEQs.

The preparation of NBs, NEQs, and NPs was performed on the Spray Dryer laboratory scale equipment (Yamato ADL-311, Santa Clara, CA, USA). The equipment was coupled to a nozzle at a setting of 0.406 mm. During the spray-drying process, all sample solutions were well diffused and thermostated using a digital hot plate stirrer (VWR VMS-C7, Corning Mexicana, Mexico). The feed rate was 1.5 mL/min, using an inlet temperature of 120 °C and an outlet temperature of 60 °C ± 5 °C with a pressure between 0.15 and 0.2 MPa. The powder obtained was kept in dark hermetic bags (Whirl-Pak^®^, Merck Mexico, Ciudad de México, Mexico) at room temperature for subsequent analysis. 

### 2.4. Characterization of NEQs

#### 2.4.1. Dynamic Light Scattering (DLS)

The average size, size distribution, and polydispersity (PDI) of the optimal inulin–soy protein nanoparticles loaded with (-)-epicatechin, and quercetin (NEQs), inulin nanoparticles loaded with (-)-epicatechin, and quercetin (NPs) and inulin–soy protein non-loaded nanoparticles (NBs) were measured by dynamic light scattering (Zetasizer DTS 1060, Malvern Instruments, Malvern, UK) and Zetasizer software (Malvern Instruments, Malvern, UK) for analysis. For the reconstitution of charged and uncharged nanoparticles, 500 µg/mL diluted in 30% ethanol and samples were sonicated (Branson 2510, Danbury, CT, USA) for 10 min and centrifugate at 10,000× *g* for 30 min at 4 °C before reading at a refractive index of 1.450 and with a dispersion optics at 173 °C (λ = 633 nm) in triplicate [21].

#### 2.4.2. Scanning Electron Microscopy (SEM)

Scanning Electron Microscopy (SEM) was performed with an EVO25 microscope (Zeiss, Cambridge, UK). Before the analysis, the samples were coated with a 4-nm layer of 50 k gold alloy in a Quorum Q150R ES (Quorum, Laughton, UK). A total of 165 nanoparticles were measured in different areas of each sample. Then a non-linear regression of Gaussian type was performed to obtain the average size of the nanoparticles.

#### 2.4.3. Fluorescence Microscopy (FM)

Nanoparticles were analyzed by fluorescence in a Zeiss Axio Imager Z1 Microscope (Carl Zeiss, Oberkochen, Deutschland) at 63× magnifications. Two fluorescence filters, Rhodamine (352–477 nm) and GFP (457–538 nm), were used, with an image magnification of 63×. These two fluorescence filters were selected because the highest fluorescence emission wavelength peaks of quercetin and (-)-epicatechin lie at 540 nm and 320 nm, respectively. Then, the colocalization of both emission spectra within the nanoparticles was analyzed. 

### 2.5. Zeta Potential

The surface charges of NBs, NPs, and NEQs were measured by Zetasizer DTS 1060 (Malvern Instruments, Malvern, UK) and Zetasizer software (Malvern Instruments, Malvern, UK). For sample reconstitution of charged and uncharged nanoparticles, 5 mg of NBs, NPs, and NEQs were diluted in 10 mL of 30% ethanol solution and the sample was sonicated for 10 min and centrifugated at 10,000× *g* for 30 min at 4 °C before reading at 37 °C in triplicate.

### 2.6. Encapsulation Efficiency (EE)

To determine the total amount of (-)-epicatechin and quercetin that were present in NEQs, and the number of flavonoids not encapsulated by the inulin matrix, the methodology proposed by Palma et al. [22] was used with slight modifications. Those modifications are described in each of the following sections. Nanoparticles of experimental design were determined by spectrophotometer (GENESYS 10S UV-Vis, Thermo Fisher Scientific, Madison, WI, USA) (λ = 320 nm) in triplicate. Optimal nanoparticles (NEQs) and NPs were measured in HPLC-DAD in triplicate as explained in Section 2.7.

#### 2.6.1. Total Flavonoids Content

NEQs (100 mg) were dispersed in 4 mL of water: ethanol: acetone (50:25:25 *v*/*v*/*v*), stirred for 1 min, ultrasonicated (Branson 2510, Danbury, CT, USA) for 20 min at room temperature, and centrifuged for 30 min at 3500 rpm. The supernatant was measured in triplicate.

#### 2.6.2. Free Flavonoids Content

NEQs (100 mg) were placed in 4 mL of ethanol and washed gently with a micropipette. The ethanol was used to extract flavonoids from the wall without breaking the polymer matrix. After the washing step, the supernatant was subjected to water: methanol: acetonitrile (45:40:15 *v*/*v*/*v*) solution plus 1% glacial acetic acid, and then it was read.

#### 2.6.3. Encapsulation Efficiency (EE) 

It was compared with a predetermined calibration curve of quercetin (R2 = 0.955) and (-)-epicatechin (R2 = 0.995), and the following formulas were used [22].
(2)$$\mathrm{EE}\;(\%)=(\frac{\mathrm{Experimental total flavonoid}-\mathrm{Surface flavonoid}}{\mathrm{Experimental total flavonoid}})\times 100$$

### 2.7. HPLC Analysis 

HPLC analysis was performed using an Agilent 1200 series pump, equipped with an autosampler and a diode array detector (DAD) (Santa Clara, CA, USA). The HPLC system was controlled by Agilent ChemStation (Agilent Technologies 2010). The NEQs solutions were filtered through 0.20 µm nylon membrane (Corning, New York, NY, USA). The concentration of each flavonoid, in each sample, was calculated from the area under the curve of its chromatographic peak using the standard curves prepared for each flavonoid on the day of the experiment. The standard deviation of each determination was always less than 5%. Detection wavelengths were 269 nm for (-)-epicatechin and 360 nm for quercetin. Other chromatographic conditions were as follows: flow rate of 0.5 mL/min, column Zorbax SB-Aq column (150 × 3.0 mm, 3.5 μm, Agilent Technologies, Santa Clara, CA, USA), oven column at 40 °C.

All flavonoids were analyzed under gradient conditions using a mobile phase mixture of acetonitrile (ACN) (A) and water/formic acid (0.2%) (B). The retention times of (-)-epicatechin and quercetin were 4.84 and 7.29 min, respectively.

### 2.8. In Vitro Flavonoid Release 

The amount of released (-)-epicatechin and quercetin were quantified using the validated HPLC method (Section 2.7) with the help of calibration curves (pH 7.4, at 37 °C, quercetin R2 = 0.955 and (-)-epicatechin R2 = 0.995) (Appendix A). All experiments were performed in triplicate.

For the quantification of released flavonoids as a function of time, 250 mg of NEQs were resuspended in PBS (pH 7.4, 100 mL) at a temperature of 37 °C for a final concentration of 2.5 mg/mL and kept under magnetic stirring at 250 rpm [22].

The concentration of flavonoids released was determined from aliquots of 1 mL collected at 0, 5, 10, 15, and 30 min. After the first 30 min, measurements were made every 30 min until 3 h of the experiment, and finally were performed every 3 h to complete 24 h. The aliquot was taken by extracting 1 mL of the analysis solution and reintroducing it into the buffer medium PBS to preserve the total volume. Data were plotted and analyzed using Origin 2022 and Excel software.

### 2.9. Mathematical Model 

To describe the physical mechanism in NEQs for flavonoids release, mathematical models such as Weibull (3), Higuchi (4), Hixson–Crowell (5), Korsmeyer and Peppas (6), and Lindner–Lippold (7) were applied.
(3)$$M=M0[1-e^{\left(-(t-T)b/a\right)}]$$
(4)Mt/M∞=k√t
(5)$${M0}^{\surd }-{Mt}^{\surd }=k_{s}t$$
(6)$$Mt/M\infty =(kt{)}^{n}$$
(7)$$Mt/M\infty =k\cdot t^{n}+b$$

The mathematical model was applied for the first 3 h of release. In the Weibull model (Equation (3)), “*a*” is the scale parameter that defines the time scale of the process (time dependence), “*t*” is the lag time before the onset of the dissolution or release process, “*t*” is time, “*M*” is accumulated fraction of flavonoid, and “*b*” describes the shape of the curve [23]. For the Higuchi model (Equation (4)), “*k*” dissolution constant, and “*Mt*/*M*∞” is the amount of flavonoid released in time t. In the Hixson–Crowell model (Equation (5)) “*ks*” represents the dissolution rate constant, Mt denotes the remaining weight of the solid at time *t*, and *M*0 is the initial weight of the solid at time *t* = 0. For the Korsmeyer–Peppas model (Equation (6)), where *Mt*/*M*∞ is a fraction of the flavonoid released at time *t*, *k* is the rate constant (having units of *t^n^*) incorporating structural and geometric characteristics of the delivery system, and *n* is the release exponent indicative of the mechanism of transport of flavonoid through the polymer matrix. Finally, in the Lindner–Lippold model (Equation (7)), “*b*” is the representation of the burst effect; “*k*” represents the rate constants, and *t* is the time. 

### 2.10. In Vitro Digestion of NEQs

The simulated gastrointestinal digestion of NEQs, NBs, and NPs in vitro was carried out according to a previous study with some modifications [24]. The digestion procedure was performed in a constant-temperature shaking incubator (VWR Model 1585) with the conditions at 37 °C and 100 rpm/min in the dark. In the first step of GIT, 200 mg of ground NEQs samples was suspended in 8 mL of water at pH 5 and supplemented with 1.5 mg/mL of α-amylase A-3176 (Sigma-Aldrich, Monterrey, Mexico) to emulate oral conditions. The pH value of the mixture was adjusted to 2.0 (using 1 M HCl) for inactivating the α-amylase and followed by a simulated gastrointestinal digestion experiment. The gastric protein hydrolysis was performed by adding 0.05 mg/mL of porcine pepsin P-7000 (Sigma-Aldrich, Monterrey, Mexico) to 40 min. Then, the pH of hydrolyzates was raised to 8 by adding a solution with 1 M NaOH to which 0.25 mg/mL of pancreatin P-7545 was added (Sigma-Aldrich, Monterrey, Mexico). The digestion was allowed to proceed for 120 min. The enzyme reaction was inactivated by placing hydrolyzates in a hot water bath adjusted to 80 °C for 2 min. After standing for 1 h, the samples were centrifuged for 10 min at 10,000× *g* and 4 °C (SL16R, ThermoScientific, Karlsruhe, Germany). The resulting supernatants were separated by filtration through a 0.20 mm sterile filter (Corning, New York, NY, USA) into 1.5 mL vials, which were immediately stored at −20 °C. The GI stability was expressed in terms of change in particle size and zeta potential during incubation in each simulated GI fluid on intervals of time using a Zetasizer DTS 1060 (Malvem Instruments, Malvern, UK) and Zetasizer software (Malvern Instruments, Malvern, UK).

### 2.11. In Vitro Cell Viability Assay

Human colorectal cancer (Caco-2), hepatocellular carcinoma (HepG2), and human dermal fibroblasts (HDFa) cell line obtained from the American Type Culture Collection (ATCC, Manassas, VA, USA) were cultured in Petri dishes with Dulbecco’s modified Eagle’s medium (DMEM) supplemented with 10% fetal bovine serum (FBS) and 1% of the commercial mixture of streptomycin and penicillin (Pen Strep Gibco©). Cells were incubated at 37 °C and 5% CO_2_. To evaluate the effect of NEQs, NBs, NPs, (-)-epicatechin, quercetin, and inulin on cell viability, cells were seeded in 96-well plates at aconcentration of 5 × 105 cells/mL and incubated for 24 h. All samples were tested based on quercetin concentration. Furthermore, inulin–soy protein non-charged nanoparticles (NBs) were used as negative control.

Measurements were made with MTS (3-(4,5-dimethylthiazol-2-yl)-5-(3-carboxymethoxyphenyl)-2-(4-sulfophenyl)-2H-tetrazolium) and PMS (phenazine methosulfate) as a stabilizer. The viability was measured according to the manufacturer’s instructions of Cell Titer 96 Aqueous One Solution Kit^®^ (Promega Corporation, Madison, WI, USA). Briefly, absorbance was measured at 490 nm on a 96-well plate reader (Synergy HT, Bio-Tek, Winooski, VT, USA). Cell viability was calculated by dividing the absorbance of the treated cells by the absorbance of the control (untreated) cells, and this ratio is expressed as a percentage.

### 2.12. Statistical Analysis

The data were expressed as mean ± SD of three independent experiments and were subjected to analysis of variance (ANOVA), and differences among means were compared by Tukey tests at *p* < 0.05. A value of *p* < 0.05 was considered significant. GraphPad Prism software 9 (GraphPad Software, San Diego, CA, USA) and Minitab software 19 (Minitab software, College, PA, USA) was used for the statistical analysis.

## 3. Results and Discussion

### 3.1. Optimization of the Spray Drying Process

The use of hybrid nanoparticles has been demonstrated to have more advantages compared to nanoparticles made of a single material. There are precedents of different combinations of polysaccharides (with protein, lipid, and synthetic biopolymer) having an advantage in retention and release characteristics of encapsulated component [25]. NEQs made from an oligosaccharide and vegetal protein matrix, inulin–soy protein, were prepared and characterized as potential nanoparticles to be supplied orally to prevent NCDs. Spray drying was chosen as the encapsulation method due to its scaling-up potential in functionalized food production. Spray drying is the most widely used technique for encapsulation on micro and nano scales [25]. It is commonly used in the food industry due to the fact that it is a clean technique that does not use solvents. It is cheap, flexible, fast, cost-effective, and easy to scale. Moreover, the spray-drying method does not affect the sensory and textural characteristics of a finished product [25]. Moreover, the equipment is used for the formulation of nanoparticles based on oligosaccharide–protein. 

In addition to their composition, the performance of nanoparticles is also related to the method and technology chosen to produce them [25]. Spray drying is a fast and cost-effective technique that is used to fabricate nanoparticles from a liquid sample using commercially available equipment [26]. This method of nanoparticle synthesis also gives the user the ability to control particle size, distribution, shape, porosity, density, and chemical composition by changing parameters such as nozzle size and spray velocity [14]. To optimize the formulation of the delivery system, a Box–Behnken design was used, with three different independent variables as reported in Table 1. The objective was to optimize particle size to a minimum, PDI, and Encapsulation Efficient (EE %), leading to a total of 17 independent experiments. The PDI surface graph showed zero inclination, and the p-value was 0.566. in this case, there are no significant model terms, which indicated the inulin % and flavonoid proportions do not affect the polydispersity index. The size of the surface, measured by SEM, showed a similar behavior to PDI, but in this case, interactions between inulin concentration (%) and flavonoid proportions ((-)-epicatechin and quercetin) affect the size of nanoparticles. In entrapment efficiency (EE %), higher concentrations of inulin and low concentrations of flavonoids (proportion 1:50) achieve a more effective encapsulation. The inulin directly correlated with the size, and EE %. Soy protein plays an important role in the stability, shape, and size of produced nanoparticles. Soy protein has been reported to form a high-protein film on the surface of the nanoparticle due to the spray-drying process of the soy protein migrates to the air droplet interface. The hot air in the dryer causes the film to turn into glassy skin [27]. 

The optimal conditions for the formulation were 4.50% of inulin, 3.50% of soy protein, and inulin:flavonoids proportion of 1:50. As a result of optimized conditions, three different fine powders were obtained: inulin–soy protein nanoparticles loaded with quercetin and (-)-epicatechin (NEQs), inulin-only nanoparticles loaded with quercetin and (-)-epicatechin (NPs) and unloaded inulin–soy protein nanoparticles (NBs). The encapsulation efficiency (EE) of both flavonoids in the loaded nanoparticles is presented in Table 2 and was significantly higher in NEQ nanoparticles (92.05–97.04%) than in NP nanoparticles (88.72–91.07%) (*p* < 0.05). Thus, a hybrid matrix was able to encapsulate higher quantities of flavonoids than a single material matrix.

### 3.2. Chemical Stability, Zeta Potential, Mean Particle Size and Flavonoid Content Analysis

Average nanoparticle size analysis through DLS showed that NEQs exhibited an increase in the size of approximately 60% when compared to NBs in an aqueous solution (Table 2). Moreover, SEM imaging revealed an average size increase of 26% for NEQs compared to NBs (Figure 1), suggesting successful encapsulation of flavonoids. The difference in measured size can be attributed to the method employed for analysis. The size of obtained nanoparticles is similar to those obtained by Li et al. [28], where they simultaneously loaded catechin and quercetin in a chitosan matrix, obtaining a 171.0 ± 2.7 nm size for blank nano-particles and 190.7 ± 2.8 nm when loaded. Sizes of less than 200 nm are desirable as they are more likely to cross the blood–brain barrier (BBB) and therefore can be used for chronic brain-associated diseases such as Alzheimer’s [29]. 

The polydispersity index (PDI) represents the intensity of light dispersed by various percentages of particles varying in size. While a PDI of 0.1 is regarded as extremely monodisperse, values of 0.1–0.4 are moderate, and 0.4 to 1.0 were highly polydisperse with multiple particle size populations [30]. The values of PDI in NBs and NEQs increment in aqueous solution in contrast to ethanol solution. In the case of NPs, the PDI almost maintained the same value. The solution where nanoparticles are dissolved, organic or aqueous, significantly influenced the final value of zeta potential, size, and PDI. The reason might be that on the surface of nanoparticles are free flavonoids, which are soluble in organic solvents. Furthermore, polymeric chains of inulin were slightly relaxed in aqueous mediums, and this usually occurs when inulin forms a hydrophilic layer [18].

The zeta potential, or electrokinetic potential, is the potential in the slipping/shear plane of a colloid particle moving in an electric field. The amount of work required to bring a unit positive charge from infinity to the surface without acceleration is known as the electric potential of a surface [30]. Zeta potential is related to pharmacokinetic properties in the body and the facility of nanoparticle phagocytosis in the bloodstream or cell [31]. The results of the zeta potential of each nanoparticle (Table 2), NEQs, NBs, and NPs were significantly different compared to each one. The NEQs’ zeta potential is the highest negative value with −18.267 ± 0.83 and −27.600 ± 4.19 in aqueous and organic solutions, respectively. The strong anionic nature of these particles was attributed to the presence of soy protein at pH 7 [32], which has a negative charge. Soy protein nanoparticles tend to have negative zeta potential. According to Renu et al., [3] the blank nanoparticle, only composed of soy protein, showed the highest negative zeta potential with −36.8 ± 1.0 mV, in comparison to curcumin-loaded soy protein nanoparticles with −34.5 ± 1.4 mV. The negative charges in nanoparticles could evade phagocytosis and induce opsonization and clearance by maintaining the surface charge as non-positive. Therefore, the delivery of nanoparticles and efficiency accumulation in tumors may improve drug efficacy [33].

NEQs encapsulation efficiency was 97.04 ± 0.01 and 92.05 ± 1.95 %, (-)-epicatechin and quercetin, respectively. The encapsulation efficiencies were higher for (-)-epicatechin than quercetin, which might be because the spatial arrangement affects the hydroxyl group’s availability to form hydrogen bonds (Table 2). Morelo et al. [34] reported the same phenomenon in inulin–soy protein microparticles loaded only with quercetin and inulin–soy protein microparticles loaded with (-)-epicatechin. Moreover, equipment efficiency in all nanoparticles was around 60%. In conclusion, the use of soy protein showed a stabilizing function in inulin as a hybrid encapsulating agent, showing a smaller size and PDI than NPs in the DLS method due to the lack of agglomeration.

SEM (Figure 1) imaging showed that all nanoparticles were spherical. NBs and NEQs presented a regular spherical shape, NBs presented the smallest size due to the lack of (-)-epicatechin and quercetin, corroborating DLS results (Table 2). On the other hand, SEM images of NPs are presented in Appendix A. NPs showed agglomeration since inulin is hygroscopic, which is one of the reasons why it was observed that the size is significantly greater both by DLS (440.93 ± 55.8 nm) and by SEM (435.5 ± 75.16 nm). NPs presented an irregular spherical shape with a rough surface, with adherence to the laminar layer and accumulation, which could be attributed to the exclusive use of inulin as an encapsulating agent. This can explain the size reported in Table 2 by DLS since the NPs presented the largest size in both aqueous and organic phases. The size of the NEQ nanoparticles was more defined as spherical and did not present agglomeration.

The average sizes shown in Figure 1c,d are different compared to those shown in Table 2, the reason being that SEM analyzes the morphological appearance of the particles, and DLS quantifies particle size distribution and estimates the molar particle mass, especially for spherical particles [35]. Additionally, SEM is a static analysis, while for DLS the particles are dissolved in an aqueous solution. Moreover, Table 1 shows heterogenous sizes in all nanoparticles and the size is the average of multiples size in the nanoparticles, which is common in spray-drying nanoparticles. According to Mlalila et al. [36], the PDI value in spray drying when showing a high value could indicate a high potential for aggregation.

In order to show that loaded nanoparticles contained both quercetin and (-)epicatechin, Fluorescence microscopy (FM) was used to show colocalization of their autofluorescence spectra within NPs (Figure 2b) and NEQs (Figure 2c), while also confirming their spherical morphology (Figure 2).

### 3.3. Effect of Temperature on the Size of NEQs

NEQs matrix are composed of around 96% inulin and the rest soy protein; NPs are 100% inulin, and both are charged with (-)-epicatechin and quercetin. NBs are only composed of inulin and soy protein in the same percentages as NEQs. The effect of temperature in aqueous mediums is important to analyze. This describes the stability and solubility of nanoparticles. Inulin is a polysaccharide which contains β-d-fructofuransonyl units, usually with an α-D-glucopyranose terminal group [17]. The temperature is related to the solubility of inulin. The inulin’s aqueous solubility is about 280 mg/mL at 80 °C and, at room temperature, it is mostly insoluble because some fractions did not dissolve at room temperature [19]. Furthermore, the solubility depends on the types of inulin, but solubility increases at higher temperatures for all different inulin polymers [37]. 

On the other hand, soy protein is constituted of two globular protein fractions: hexameric glycinin, and trimeric β-conglycinin. Soy protein particles form a hydrophobic core structure with basic polypeptides and β subunits interactions and acidic polypeptides, α′ and α subunits, located on the outside of the core, forming the hydrophilic shell [38]. The solubility of soy protein is influenced by multiple factors, such as dissolution temperature, pH, and ionic strength [39].

Figure 3 shows the relation between temperature and the average size of nanoparticles, in all nanoparticles, the size increased at a low temperature, due to inulin forming most of the nanoparticle matrix compared to soy protein. According to Jiménez-Sánchez et al. [40], they observed that decreasing the temperature resulted in an inulin nanoparticle size increase. Charoenwongpaiboon et al. [41] also showed similar behavior. The sizes were dependent on the temperature, with dimensions between 95.9 and 115.7 nm for temperatures between 50 and 40 °C, respectively, in inulin nanoparticles. The explanation of the indirect relation between size and temperature is due to the interaction between the inulin with the medium and the temperature. At higher temperatures, the inulin dissolves in the medium, which generates a smaller size in the nanoparticle, by reducing its encapsulating matrix.

### 3.4. In Vitro Drug Release

The drug release in NEQs and NPs is different in each case at the same conditions, 37 °C, and a PBS solution at pH 7.4. The results were deeply analyzed and compared with the parameters referenced for each mathematical model (Appendix A) and have been presented in Table 3 and Table 4. Figure 3b shows a burst effect in NPs in both flavonoids. The cumulative drug release drastically increased in the first 25 min. Furthermore, the burst effect can be proven with variable b in Lindner and Lippold’s model, with positive values in variable b, 2.19 ± 3.90 and 11.99 ± 10.10, (-)-epicatechin and quercetin respectively (Table 3 and Table 4). The Weibull function (Equation (3)) was used in NPs to present the best fit, with a R^2^ value at 0.917 and 0.972, (-)-epicatechin and quercetin respectively (Table 3 and Table 4). The mathematical model indicates a complex release mechanism. Moreover, the second equation that best fits was the Korsmeyer–Peppas model (Table 3 and Table 4), which describes sphere geometry with anomalous transport. The geometry form was confirmed by SEM images (Figure 1). 

On the other hand, NEQs (Figure 3c) did not present a burst effect and the b values were negative. The Korsmeyer and Peppas model was the best-fit model in (-)-epicatechin and the second-best in quercetin. Moreover, n values indicate a sphere geometry with an anomalous transport. The Weibull model showed a good fit with an R2 of 0.966 and 0.987, (-)-epicatechin and quercetin, respectively. The b variable of the Weibull equation indicates a diffusion in a fractal or disordered substrate in both flavonoids (Table 3 and Table 4).

The different drug release in both nanoparticles is a consequence of the addition of soy protein in the matrix of nanoparticles. Shivhare et al. [42], showed similar behavior in inulin–peptide nanoparticles loaded with ornidazole. In the first hours, the rapid burst release effect was present, and more than 90% or more ornidazole was released from the nanoparticle at pH 7.4 after 24 h. Another study showed that soy protein had a significant effect on the flavonoid release ((-)-epicatechin and quercetin microparticles) with a positive impact on the flavonoid release behavior in hexane [36]. 

### 3.5. Variation in Physicochemical Properties of NEQs, NPs, and NBs under Simulated Gastrointestinal Digestion In Vitro

#### 3.5.1. (-)-Epicatechin Release and Quercetin Characteristics

The release property of (-)-epicatechin and quercetin from NEQs and NPs were analyzed under simulated gastrointestinal digestion in vitro (Figure 4a,b). During the first 30 min, the gastrointestinal conditions correspond to the initial GIT process, chewing, with α-amylase and pH 5. (-)-epicatechin, unlike quercetin, was shown to start its release from NEQs at less than 10%, which might be that a small amount of (-)-epicatechin attached to the surface, more in the matrix with only inulin (NPs) compared to inulin–soy protein matrix (NEQs). In pH 4.5, the gastric phase with porcine pepsin was simulated, and results suggested that inulin was hydrolyzed (in a low amount) by acidic environments in both nanoparticle NPs and NEQs (Figure 4a,b). Inulin hydrolysis in the gastric phase has been previously reported [19]. However, the addition of soy protein made the release of both flavonoids significantly slower in NEQs than in NPs. As shown in Figure 4a, at 75–80 min the release of (-)-epicatechin was twofold higher in NPs than in EQs (*p* < 0.05) in consequence. Figure 4b showed that during the gastric phase, the delivery of quercetin was significantly greater in NPs than in NEQs (*p* < 0.05). At the end of the gastric phase, quercetin reached a release of 88% in NPs, while in NEQs it was 61%. Those results showed that NEQs were able to better retain flavonoids during the gastric phase and therefore more effective in bringing a higher concentration of flavonoids to the intestinal phase. Finally, during the small intestine phase, because of the dramatic difference in the digestive environment, the (-)-epicatechin was rapidly and sharply liberated from the nanoparticles in both cases, NEQs and NPs (Figure 4a). NEQs showed a more controlled release rate of quercetin compared to NPs. Therefore, the cumulative concentration of quercetin was significantly greater in NPs (89.8 ± 3.7%) compared to those in NEQs (74.9 ± 2.4%) at 120 min (Figure 4b). After exposure to the small intestinal fluid, the maximum cumulative release rates of NEQs were 88.5 ± 3.9 and 91 ± 2.6% for (-)-epicatechin and quercetin, respectively. NPs showed a final cumulative release concentration of 92.2 ± 4.6 and 90 ± 8.5% for (-)-epicatechin and quercetin, respectively. In both cases, NEQs and NPs showed no significant differences in the final cumulative release of both flavonoids. However, the rate of release of the compounds did differ by the content of soy protein in NEQs, making NEQs more stable in the gastric phase and having a greater release of flavonoids in the intestinal phase compared to NPs (Figure 4a,b). Results suggested that the addition of soy protein in matrix encapsulation (NEQs) shows a better protective capacity against extreme conditions across the gastrointestinal tract. 

#### 3.5.2. Zeta Potential

The zeta potential of NBs, NEQs, and NPs was measured at a different phase of GIT in vitro. As shown in Figure 4c, the charges of nanoparticles experienced variation from negative in the mouth stage, strongly positive in gastric fluid, and negative charge in the final stage, intestinal environment. During exposure to extremely low pH and high ionic strength in gastric fluid, the zeta potential value of both nanoparticles presented an increased tendency in a cationic potential. When the process stepped into small intestinal fluid, the zeta potential of NBs, NEQs, and NPs decreased sharply, due to the deprotonation of polysaccharides occurring, leading to the decrease of negative zeta potentials [43]. During the intestinal phase, the negative value of the zeta potential of all nanoparticles started to increase slowly and stabilized. Cheng et al. [44] reported a decreased value of potential zeta of soy protein nanoparticles loaded with folic acid from −18 mV to −39 mV, when the pH value increased from 5.0 to 9.0. The isoelectric point (IP) of soy protein is influenced by zeta potential. When the value pH of the solution is higher than IP, the particle had a negative surface, and the opposite when it is lower than IP. Zeta potential commonly is used as a parameter for relating it to the stability of dispersed systems. The reported zeta potential threshold for stable systems has been considered within ±20 to ±30 mV [45]. According to this information and considering the zeta potential of present loaded nanoparticles presented in Figure 4c and Table 2, are moderately stable to highly stable depending on the pH value of the system.

#### 3.5.3. Average Particle Size

The changes in particle sizes of NBs, NEQs, and NPs during GIT in vitro were investigated, as shown in Figure 4d. In the first stage of GIT, the mouth, the average particle diameter of nanoparticles was maintained. In the early gastric phase, the size of nanoparticles dramatically dropped. The reason might be inulin; the percentages in NBs and NEQs were around 97% and 100% in NPs. The inulin cannot be digested by human intestinal digestive enzymes, but in highly acidic environments β-(2–1) bonds between the fructose units may be partially hydrolyzed [46]. Moreover, the (-)-epicatechin and quercetin release in this stage affects the size of nanoparticles. Interestingly, when the digestion was stepped into the small intestinal stage, the particle size of NPs decreased sharply, in contrast, NEQs and NBs maintained the average particle size. The reason might be due to variations in digestive conditions changing the structural characteristic of nanoparticles and the release rates of (-)-epicatechin and quercetin.

### 3.6. Cell Viability

Considering the encapsulation efficiency, drug release, average size, zeta potential, and performance in GIT, inulin–soy protein loaded with (-)-epicatechin, and quercetin (NEQs) was selected to investigate the cytotoxicity activities on the colon (Caco-2) and the liver (HepG2) cells lines (Figure 5). The use of this type of cell line related to the digestive system was the purpose of nanoparticles. They are developed for oral treatment, such as a nutraceutical, aiming at the prevent the development of NCDs. In addition, the behavior of nanoparticles in nontumoral cell lines such as human dermal fibroblast (HDFa) was studied. Inulin–soy protein unloaded was used as a blank nanoparticle, and the components of the nanoparticle in pure form (inulin, (-)-epicatechin, and quercetin) to validate their cytotoxic effect. An MTS assay was used to test the cell viability. The results suggested that NEQ nanoparticles promoted the internalization of (-)-epicatechin and quercetin in Caco-2 cells, contributing to the better cytotoxicity effect.

Treatment with NEQs resulted in a significant decrease in cell viability in the Caco-2 cell line in contrast to the control cell at concentrations under 1.18 µg/mL (*p* < 0.05) (Figure 5a). Free inulin presented a reduction of viability of around 20% at the highest concentration, but the lowest (1.18 and 0.59 µg/mL) showed proliferation of cells. There is a significant difference between free (-)-epicatechin and quercetin effects in cell viability compared to NEQs at 4.75 to 1.18 µg/mL. NEQs exhibited a stronger inhibitory effect on the viabilities of Caco-2 cells, with the IC50 value of 4.09 µg/mL, suggesting that the antiproliferative activity of (-)-epicatechin and quercetin enhanced by encapsulation and simultaneous loaded with inulin–soy protein matrix. In addition, the internalization of NEQs into a cell could be due to the size of nanoparticles 280.17 ± 13.42 nm in water. 

As shown in Figure 5b, the HepG2 cell line, NEQs, and NBs showed a slightly increased viability in the lowest concentrations (1.18–0.59 µg/mL). In contrast, free (-)-epicatechin showed a decreased cell viability of around 20% at 1.18 µg/mL, and quercetin a reduction of cell viability of around 30% at 4.75 µg/mL. Inulin did not present an inhibitory effect on cell viability. In both cell lines, Caco-2 and HepG2, it was observed that no anti-proliferation effect was observed at all concentrations regarding NBs (unloaded nanoparticles), and the cell viability remained above 80% in all treatments. 

HDFa is a human dermal fibroblast cell line, as shown in Figure 5c, NEQs showed greater than 90% cell viability, on average, at all tested doses (0.59–4.75 µg/mL). Free (-)-epicatechin and quercetin did not show a significant difference with NEQs in all concentrations. On the other hand, NBs and free inulin did not show a toxic effect. Nanoparticles based on polysaccharides did not show cytotoxicity in peripheral blood mononuclear cells (PBMCs) according to Jiménez-Sánchez et al. [40]. This suggests that NEQs have significant cytotoxicity effects against Caco-2 and minimal effects against HepG2 and normal HDFa cells.

## 4. Conclusions

Inulin–soy protein was utilized as a carrier to simultaneously load (-)-epicatechin and quercetin, which was confirmed through Fluorescence Microscopy. The Box–Behnken design was employed to optimize the synthesis formulation of nanoparticles. The optimized formulation involved an inulin concentration of 4.5% (*w*/*v*), soy protein concentration of 3% (*w*/*v*), and an inulin:flavonoids proportion of 1:50 ((-)-epicatechin and quercetin proportion of 1:1). The size of the inulin–soy protein nanoparticles loaded with (-)-epicatechin and quercetin (NEQs) was 280.17 ± 13.42 nm in water and 170.43 ± 16.51 nm in ethanol. The zeta potential of NEQs was −27.600 ± 4.19 mV in ethanol and −18.267 ± 0.83 mV in water. The soy protein in NEQs had a positive impact on drug release, did not exhibit a burst effect, and showed progressive release during the first 3 h, in contrast to inulin nanoparticles (NBs), which displayed a burst effect with approximately 80% of (-)-epicatechin and quercetin released in the medium during the first hour. In vitro digestion experiments demonstrated that soy protein had a protective effect. Additionally, NEQs released (-)-epicatechin and quercetin in the intestinal phase, exhibiting a negative zeta potential that might enhance the internalization of nanoparticles in gastrointestinal cells. Nanoencapsulation of (-)-epicatechin and quercetin into hybrid encapsulating agents, inulin–soy protein, enhances their bioavailability in the gastrointestinal tract (GIT).

The MTS assay revealed significant cytotoxicity of NEQs up to a concentration with an IC50 value of 4.09 µg/mL in the Caco-2 cell line, indicating that the internalization of NEQs led to the potential toxic effect of (-)-epicatechin and quercetin in carcinogenic cells. In the case of HepG2 and HDFa, the cell viability was more than 70%.

These results provide crucial information for designing and fabricating biopolymeric nanoparticle delivery systems for two flavonoids. NEQs can be used in the food industry by spray-dryer equipment as a part of the future development of functional foods to prevent non-communicable diseases (NCDs). Further research is required to assess the antioxidant effect and storage stability and to test the performance and efficacy of NEQs in vivo using an animal model.

## Figures and Tables

**Figure 1 nanomaterials-13-01615-f001:**
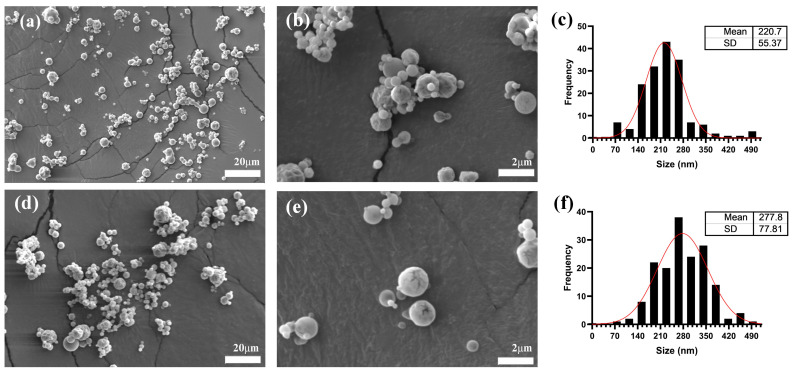
Nanoparticles size characterization by Scanning Electron Microscopy (SEM). (**a**,**b**) show images of unloaded hybrid Inulin–Soy Protein nanoparticles (NBs). (**d**,**e**) show hybrid Inulin–Soy Protein nanoparticles loaded with flavonoids ((-)-Epicatechin and Quercetin) (NEQs). (**c**,**f**) show average particle size distribution histograms for NBs and NEQs.

**Figure 2 nanomaterials-13-01615-f002:**
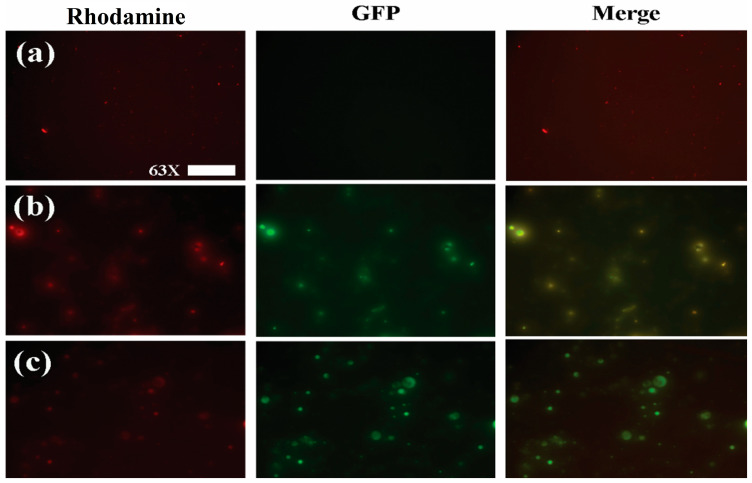
Fluorescence microscopy images of nanoparticle formulations imaged with a Rhodamine (left column) and GFP (central column) emission filters. A digital merge of both channels is shown in the right column in order to show colocalization of flavonoid fluorescence within nanoparticles. (**a**) Unloaded Hybrid Inulin–Soy Protein nanoparticles (NBs) (**b**) Inulin nanoparticles loaded with (-)-Epicatechin and Quercetin (NPs). (**c**) Hybrid Inulin–soy protein nanoparticles loaded with (-)-Epicatechin and Quercetin (NEQs). Figures were taken with 63× magnification, and the scale bar presented 50 µm.

**Figure 3 nanomaterials-13-01615-f003:**
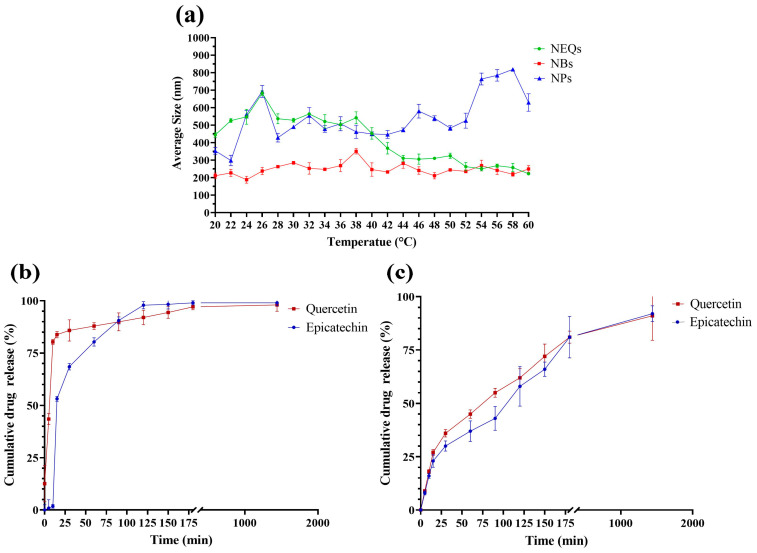
(**a**): Relation between Temperature (°C) and Average Size (nm) in a range of 20 to 60 °C in an aqueous solution (5 mg/mL), Inulin–soy protein nanoparticles loaded with (-)-Epicatechin and quercetin (NEQs, black line), Inulin–soy protein nanoparticles non-loaded (NBs, Grey line) and Inulin nanoparticles loaded with (-)-Epicatechin and Quercetin (NPs, gold line). (**b**): In vitro drug release of Inulin nanoparticles loaded with (-)-Epicatechin and Quercetin (NPs), (-)-Epicatechin (Black line) and Quercetin (Grey line) release kinetics at 7.4 pH and 37 °C at 1440 min (24 h). (**c**): In vitro drug release of Inulin–soy protein nanoparticles loaded with (-)-Epicatechin and Quercetin (NEQs), (-)-Epicatechin (Black line) and Quercetin (Grey line) release kinetics at 7.4 pH and 37 °C at 1440 min (24 h).

**Figure 4 nanomaterials-13-01615-f004:**
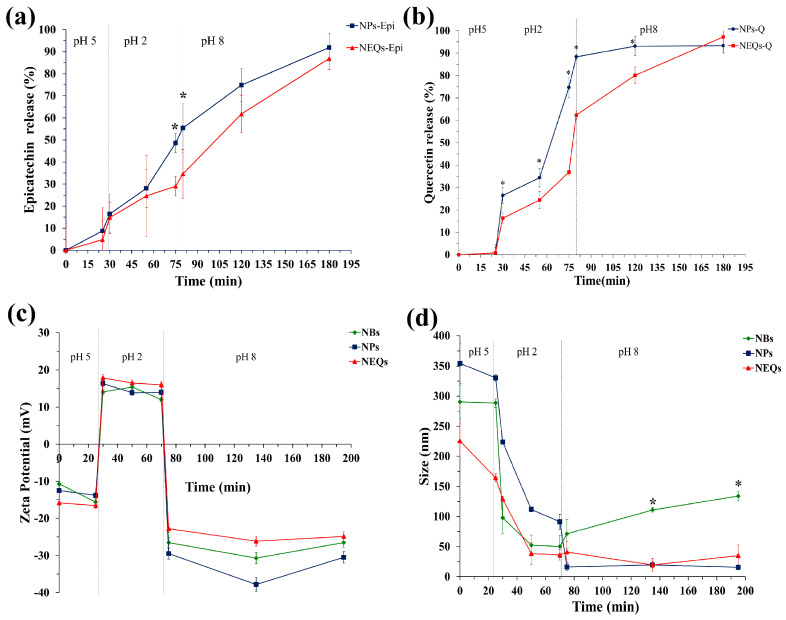
The simulated gastrointestinal digestion of inulin–soy protein nanoparticles loaded with (-)− Epicatechin and Quercetin (NEQs), inulin nanoparticles loaded with (-)− Epicatechin and Quercetin (NPs) and Inulin–soy protein nanoparticles non-loaded (NBs) prepared by different condition in vitro. (**a**) the release behaviors of (-)− Epicatechin from NEQs and NPs; (**b**) the release behaviors of Quercetin from NEQs and NPs; (**c**) variation in zeta potential from NEQs, NPs and NBs. (**d**) variation in average particle size, from NEQs, NPs and NBs. * Represents significant differences between treatments in the same time (*p* < 0.05).

**Figure 5 nanomaterials-13-01615-f005:**
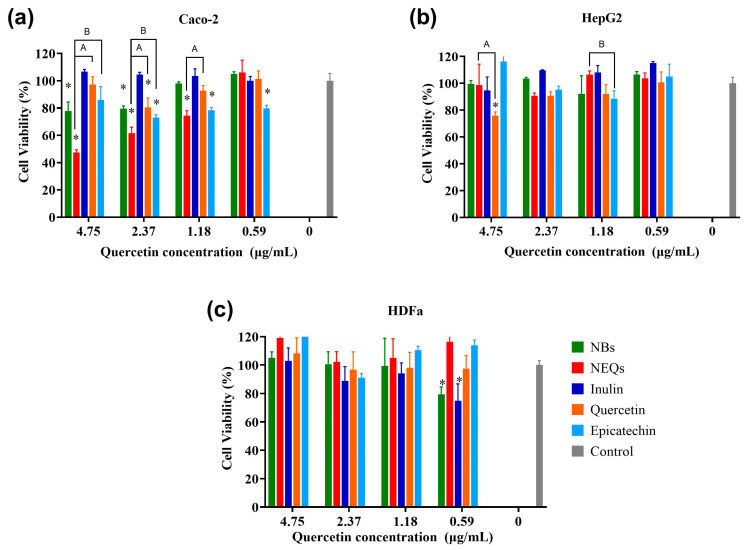
Cell viability and effect of nanoparticles. (**a**): Cell viability in human colorectal cancer cells (Caco-2). (**b**): Cell viability in hepatocellular carcinoma (HepG2). (**c**): Cell viability in human dermal fibroblast cell line (HDFa). Cell viability (%) of inulin–soy protein nanoparticles loaded with (-)-Epicatechin and Quercetin (NEQs), inulin–soy protein non-charged nanoparticles (NBs), (-)-Epicatechin, Quercetin and inulin at different concentrations (4.75 to 0.59 µg/mL) after 24 h, determined using the MTS assay. Values are expressed as mean ±SD. Controls are untreated cells. Significant differences among treatments in the cell lines were determined based on Dunnett multiple comparison test (*p* ≤ 0.05) where * control vs. all treatments. Significant differences were determined based on the Student’s t-distribution (*p* ≤ 0.05) where ^A^ represents NEQs vs. Quercetin and ^Β^ represents NEQs vs. (-)-Epicatechin.

**Table 1 nanomaterials-13-01615-t001:** Experimental design Box–Behnken. Independent variables.

Variable	Low Point	Central Point	High Point
Inulin concentration (*w*/*v* %)	3	6	9
Soy protein concentration (*w*/*v* %)	0	12.5	50
Flavonoids proportion	1:50	1:35	1:20

**Table 2 nanomaterials-13-01615-t002:** Zeta Potential, Polydispersity Index (PDI), Average size, Encapsulation efficiency (%) and equipment efficiency % (EE) of Inulin–soy protein nanoparticles loaded with (-)-Epicatechin and Quercetin (NEQs), Inulin nanoparticles loaded with (-)-Epicatechin and Quercetin (NPs) and non-charged nanoparticles (NBs) at 37 °C.

	Organic Phase (Ethanol)			Aqueous Phase (H_2_O)			Flavonoids	Encapsulation Efficiency %
	Zeta Potential (mV)	Size (nm)	PDI	Zeta Potential (mV)	Size (nm)	PDI		
NBs	−12.567 ± 1.42 ^a,^*	138.10 ± 19.48 ^b^	0.280 ± 0.02 ^a,^*	0.676 ± 1.55 ^a,^*	179.95 ± 21.71 ^b^	1.00 ± 0.05 ^a,^*	Epi	−
							Qer	−
NEQs	−27.600 ± 4.19 ^c,^*	170.43 ± 16.51 ^b,^*	0.336 ± 0.06 ^a^	−18.267 ± 0.83 ^c,^*	280.17 ± 13.42 ^c,^*	0.805 ± 0.02 ^c^	Epi	97.04 ± 0.01
							Qer	92.05 ± 1.95
NPs	−18.967 ± 2.08 ^b,^*	421.90 ± 10.32 ^a^	0.673 ± 0.01 ^a^	−14.967 ± 0.81 ^b,^*	440.93 ± 55.8 ^a^	0.643 ± 0.06 ^b^	Epi	91.07 ± 2.81
							Quer	88.72 ± 0.89

Epi = (-)-Epicatechin, Quer = Quercetin. Means ± SD in the same column with different letters are significantly different (*p* < 0.05). * Represents significant difference between mediums and same parameters and nanoparticles.

**Table 3 nanomaterials-13-01615-t003:** Mathematical models to prediction of kinetic release of quercetin and in the first 3 h of Inulin–soy protein nanoparticles loaded with quercetin (NEQs) and inulin nanoparticles loaded with epicatechin and quercetin (NPs).

Mathematical Model			Epicatechin			
Higuchi	**NEQs**		**NPs**			
	**k**	**R^2^**	**k**		**R^2^**	
	5.37 ± 0.17	0.969	8.59 ± 0.64		0.851	
Weibull	**b**	**R^2^**	**b**		**R^2^**	
	0.62 ± 0.32	0.966	1.55 ± 0.63		0.917	
Hixson–Crowell	k	**R^2^**	k		**R^2^**	
	0.47 ± 0.03	0.873	0.73 ± 0.10		0.558	
Korsmeyer and Peppas	**k**	**n**	**R^2^**	**k**	**n**	**R^2^**
	9.70 ± 5.97	0.58 ± 0.05	0.974	11.33 ± 5.54	0.44 ± 0.10	0.839
Lidner–Lippold	**b**	**n**	**R^2^**	**b**	**n**	**R^2^**
	−8.88 ± 17.34	0.62 ± 0.09	0.971	2.19 ± 3.90	0.39 ± 0.13	0.825

**Table 4 nanomaterials-13-01615-t004:** Mathematical models to predict of kinetic release of quercetin and in the first 3 h of Inulin–soy protein nanoparticles loaded with epicatechin (NEQs) and inulin nanoparticles loaded with epicatechin and quercetin (NPs).

Mathematical Model			Quercetin			
Higuchi	**NEQs**		**NPs**			
	**k**	**R^2^**	**k**		**R^2^**	
	5.90 ± 0.10	0.991	9.31 ± 1.18		0.247	
Weibull	**b**	**R^2^**	**b**		**R^2^**	
	0.48 ± 0.15	0.987	1.79 ± 0.38		0.972	
Hixson–Crowell	**k**	**R^2^**	**k**		**R^2^**	
	0.51 ± 0.04	0.784	0.74 ± 0.16		0.283	
Korsmeyer and Peppas	**k**	**n**	**R^2^**	**k**	**n**	**R^2^**
	6.07 ± 0.76	0.49 ± 0.03	0.990	52.93 ± 8.18	0.12 ± 0.04	0.853
Lidner–Lippold	**b**	**n**	**R^2^**	**b**	**n**	**R^2^**
	−0.71 ± 2.75	0.48 ± 0.04	0.988	11.99 ± 10.10	0.14 ± 0.05	0.861

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
