# Peer review of "Novel Hybrid Inulin–Soy Protein Nanoparticles Simultaneously Loaded with (-)-Epicatechin and Quercetin and Their In Vitro Evaluation"

_nanomaterials, 2023, doi:10.3390/nano13101615_

Round 1
Reviewer 1 Report
In this paper, authors investigate the formulation and characteristics of hybrid inulin-soy nanoparticles loaded with (-)-Epicatechin and Quercetin (NEQs) to improve the bioavailability of these flavonoids in the human body for their potential therapeutic application.
The study is interesting, exhaustive and generally well described and carried out. I consider also that the conclusions are correct according to the results. Nevertheless, I think there are parts that can be improved. Please, take these comments into consideration:
- You have estimated sizes and other parameters using DLS, and though DLS is very useful to analyze different particle parameters, it is not a very reliable technique in general to estimate particle sizes, so they should at least be compared using another technique. It could be transmission electron microscopy (TEM) although perhaps it would be more appropriate for metallic and smaller particles. In this case you can use SEM not only to provide the images showed in Figure 1, but also to do a quantitative analysis through graphic distribution and their Gaussian fittings of the at least, 100 particles from different areas of each sample, and them compare the results between the different samples and also with the results provided by DLS. These distributions could be at the supplementary material.
Another optional suggestion is the use of atomic force microscopy (AFM), but in this case AFM is suitable to analyze the morphology details of the particles in addition to analyze the average sizes. If you have the opportunity to introduce AFM images, with the advantage that can be developed in buffer, so it is more realistic regarding the future applications of the particles, it could be nice. And given the relationship found in your paper between size and temperature, AFM could also provide the possibility of analyzing size and morphology at 37ºC using a closed liquid cell. You have an example of TEM and AFM analysis of particles in doi:10.1088/0957-4484/21/46/465707.
- Implement, at the same time, Figure 1 with insets showing zoomed images where individual particles look bigger.
- What does “6179” mean in line 105, is it a strain?
-Show calibration curves in encapsulation efficiency, because you cite them but do not show them, and they must be performed with the same batches of samples.
Regarding the style:
- Delete all “we”. They are not elegant.
- Sometimes you use “rodamin” and other “rodamina”, homogenize.
- Homogenize also “a, b, c…” or “A,B,C…” in all the figures according to the instructions.
- In Figure 2, you have two “b)” and any “c)”.
- In Figure 3 caption, there is a mixture of letter types.
- Please, increase letter size in the graphics of Figures 3,4 and 5.
I am not an expert in this language, but in general, I see the correct writing, although I think that many expressions and paragraphs can be improved. I have also seen quite a few errors due to not having reviewed the version sent by the authors well.
Author Response
Dear Editor
I am submitting online the manuscript entitled "Novel hybrid Inulin-Soy Protein nanoparticles simultaneously loaded with (-)- Epicatechin and Quercetin and their in-vitro evaluation" by Ayala-Fuentes et al. for your consideration and possible publication in Nanomaterials (ISSN 2079-4991).
Our research investigates the development of hybrid Inulin-Soy Protein nanoparticles that are simultaneously loaded with (-)- Epicatechin and Quercetin, and their in-vitro evaluation. We believe that our findings have significant implications for the field of nanomaterials.
We have responded in detail to all the comments of revisors, and we want to thank the given observations to improve our manuscript.
The response to each observation is the following:
REVIEWER 1
In this paper, authors investigate the formulation and characteristics of hybrid inulin-soy nanoparticles loaded with (-)-Epicatechin and Quercetin (NEQs) to improve the bioavailability of these flavonoids in the human body for their potential therapeutic application.
The study is interesting, exhaustive and generally well described and carried out. I consider also that the conclusions are correct according to the results. Nevertheless, I think there are parts that can be improved. Please, take these comments into consideration:
- You have estimated sizes and other parameters using DLS, and though DLS is very useful to analyze different particle parameters, it is not a very reliable technique in general to estimate particle sizes, so they should at least be compared using another technique. It could be transmission electron microscopy (TEM) although perhaps it would be more appropriate for metallic and smaller particles. In this case you can use SEM not only to provide the images showed in Figure 1, but also to do a quantitative analysis through graphic distribution and their Gaussian fittings of the at least, 100 particles from different areas of each sample, and them compare the results between the different samples and also with the results provided by DLS. These distributions could be at the supplementary material.
Thank you for your suggestion. We have already included a quantitative analysis through graphic distribution and their Gaussian fittings of 165 particles from different areas of each sample. The procedure was included in materials and methods in lines 155-157. Figure 1 was modified to include the results in line 331. Also, results were presented and discussed in lines 317-320 and 357-362.
Another optional suggestion is the use of atomic force microscopy (AFM), but in this case AFM is suitable to analyze the morphology details of the particles in addition to analyze the average sizes. If you have the opportunity to introduce AFM images, with the advantage that can be developed in buffer, so it is more realistic regarding the future applications of the particles, it could be nice. And given the relationship found in your paper between size and temperature, AFM could also provide the possibility of analyzing size and morphology at 37ºC using a closed liquid cell. You have an example of TEM and AFM analysis of particles in doi:10.1088/0957-4484/21/46/465707.
Thank you for this optional suggestion, unfortunately we have no access to atomic force microscopy (AFM), However with hope the changes made in Figure 1 explain better the average size analysis.
- Implement, at the same time, Figure 1 with insets showing zoomed images where individual particles look bigger.
Thank you for the suggestion, we have already added to Figure 1 zoomed images of each sample. Figure 1 was modified in line 331.
- What does “6179” mean in line 105, is it a strain?
Thank you for the observation, it was the catalog number of the Organic Inulin from Agave powder, because this is not necessary in the manuscript, we have deleted it.
-Show calibration curves in encapsulation efficiency, because you cite them but do not show them, and they must be performed with the same batches of samples.
Thank you for the suggestion. We consider that this information is a supplementary information, but as you notice it is cited on the manuscript, so we added as a supplementary Figure (Supplementary Figure 1) and cited in line 206
Regarding the style:
- Delete all “we”. They are not elegant.
Thank you for the suggestion, we have modified “we” in lines 50, 86, 87, 162, and 447
- Sometimes you use “rodamin” and other “rodamina”, homogenize.
Thank you for the observation. We have already changed “rodamina” for “rodamin” in lines 161, and 340-343.
- Homogenize also “a, b, c…” or “A,B,C…” in all the figures according to the instructions.
Thank you for the observation, the letter in figures were homogenized in the whole manuscript as: (a), (b), (c) … according to the instructions.
- In Figure 2, you have two “b)” and any “c)”.
Thank you for the observation, we have already corrected the second (b) for (c).
- In Figure 3 caption, there is a mixture of letter types.
Thank you for the observation, the letter in figures were homogenized in the whole manuscript as: (a), (b), (c) … according to the instructions.
- Please, increase letter size in the graphics of Figures 3,4 and 5.
Thank you for your suggestions, we have increased the letter size in Figures 3, 4 and 5.
We would like to thank all comments, suggestions, and comments made by the three reviewers, as well as your time and consideration.
We believe that our manuscript would be of interest to the readership of your journal, and we look forward to your favorable consideration. Thank you for your time and attention.

Reviewer 2 Report
The authors have already published the following manuscript:
Ayala-Fuentes, J.C.; Gallegos-Granados, M.Z.; Villarreal-Gómez, L.J.; Antunes-Ricardo, M.; Grande, D.; Chavez-Santoscoy, R.A. Optimization of the Synthesis of Natural Polymeric Nanoparticles of Inulin Loaded with Quercetin: Characterization and Cytotoxicity Effect. Pharmaceutics 2022, 14, 888. https://doi.org/10.3390/pharmaceutics14050888
Some methods are presented in this articles.
The authors must pointed the novelty and the differences between the two articles.
The text from the figures are not visible.
Author Response
Dear Editor
I am submitting online the manuscript entitled "Novel hybrid Inulin-Soy Protein nanoparticles simultaneously loaded with (-)- Epicatechin and Quercetin and their in-vitro evaluation" by Ayala-Fuentes et al. for your consideration and possible publication in Nanomaterials (ISSN 2079-4991).
Our research investigates the development of hybrid Inulin-Soy Protein nanoparticles that are simultaneously loaded with (-)- Epicatechin and Quercetin, and their in-vitro evaluation. We believe that our findings have significant implications for the field of nanomaterials.
We have responded in detail to all the comments of revisors, and we want to thank the given observations to improve our manuscript.
The response to each observation is the following:
REVIEWER 2
The authors have already published the following manuscript:
Ayala-Fuentes, J.C.; Gallegos-Granados, M.Z.; Villarreal-Gómez, L.J.; Antunes-Ricardo, M.; Grande, D.; Chavez-Santoscoy, R.A. Optimization of the Synthesis of Natural Polymeric Nanoparticles of Inulin Loaded with Quercetin: Characterization and Cytotoxicity Effect. Pharmaceutics 2022, 14, 888. https://doi.org/10.3390/pharmaceutics14050888
Some methods are presented in this articles.
Yes, an earlier article has been published, in which the nanoencapusulation of only 1 flavonoid was optimized: Quercetin. This previous article is precisely the antecedent to the present original manuscript. That is the reason we cited in lines 280-285.
The authors must pointed the novelty and the differences between the two articles.
The main differences between the articles are described in the following table. It is presented in table form to make very clear the novelty and the objective of each article.
|
Article |
Wall materials |
Encapsulated compound |
Novely |
|
Ayala-Fuentes, et al., 2022 |
inulin |
Quercetin |
Encapsulation of a flavonoid in an FDA-approved polymer as an ingredient in food. |
|
Present manuscript |
1. Inulin and Soy Protein 2. Inulin |
Quercetin and Epicatechin |
Encapsulation of two flavonoids in the same nanoparticle in a hybrid wall material that can be used as food ingredient. |
The text from the figures is not visible.
Thank you for the observation, we have already increased the letter size in Figures 3, 4 and 5. Figure 1 was modified to make clarify the shown information.
We would like to thank all comments, suggestions, and comments made by the three reviewers, as well as your time and consideration.
We believe that our manuscript would be of interest to the readership of your journal, and we look forward to your favorable consideration. Thank you for your time and attention.

Reviewer 3 Report
Dear Authors,
your manuscript "Novel hybrid Inulin-Soy Protein nanoparticles simultaneously loaded with (-)- Epicatechin and Quercetin and their in-vitro evaluation" needs to be revised on several points.
1) Show the data obtained with all three NPs;
2) in the SEM images (figure 1): Include the micrograph of the NBs.
3) in the IF images (Figures 2): better explain the selection of the two filters. Indicate the emission wavelengths of Epicatechin and Quercetin;
4) in Figures 2 b and c the signal is very weak. Which substance are you referring to?
5) cell viability test: perform the test at 24, 48 and 72 hrs using the three NPs. This test is essential in order to have an indication of the biocompatibility of the nanocarriers that are expected to be used to deliver the drugs.
Finally I also invite the authors to well presenting their results, in a order mode.
The experimetal plan is very interesting but unfortunately cannot be accepted in this form.
Sincerely
Author Response
Dear Editor
I am submitting online the manuscript entitled "Novel hybrid Inulin-Soy Protein nanoparticles simultaneously loaded with (-)- Epicatechin and Quercetin and their in-vitro evaluation" by Ayala-Fuentes et al. for your consideration and possible publication in Nanomaterials (ISSN 2079-4991).
Our research investigates the development of hybrid Inulin-Soy Protein nanoparticles that are simultaneously loaded with (-)- Epicatechin and Quercetin, and their in-vitro evaluation. We believe that our findings have significant implications for the field of nanomaterials.
We have responded in detail to all the comments of revisors, and we want to thank the given observations to improve our manuscript.
The response to each observation is the following:
REVIEWER 3
your manuscript "Novel hybrid Inulin-Soy Protein nanoparticles simultaneously loaded with (-)- Epicatechin and Quercetin and their in-vitro evaluation" needs to be revised on several points.
1) Show the data obtained with all three NPs;
The information of the 3 obtained nanoparticles is described in Table 2. NBs (Nanoparticles made with inulin and soy protein, but without flavonoids) have the reports of flavonoids because we used NBs as Control nanoparticles.
2) in the SEM images (figure 1): Include the micrograph of the NBs.
NBs nanoparticles, which are hybrid nanoparticles made with inulin and soy protein but without flavonoids, are included in Figure 1. However, we did not include NPs nanoparticles (inulin nanoparticles loaded with both flavonoids), because as we discussed in lines 278-286 and 303-308 the hybrid nanoparticle (made with inulin and soy protein) showed significantly greater encapsulation efficiency of both flavonoids. Another reason was that the size of NPs was significantly greater than NBs and NEQs and presented agglomeration. The information was described in Table 2 and discussed in lines 333-336.
For your suggestion, we included SEM images as a supplementary Figure, because we want to highlight that the best formulation that we obtain to nanoencapsulate both flavonoids is the one with a hybrid wall material, as is described in the tittle and throughout all the manuscript. We referenced the Figure in lines 327-329.
3) in the IF images (Figures 2): better explain the selection of the two filters. Indicate the emission wavelengths of Epicatechin and Quercetin;
Thank you for the suggestion. We have already explained better the selection of the two filters in Materials and Methods section lines 163-165.
4) in Figures 2 b and c the signal is very weak. Which substance are you referring to?
Thank you for the observation, in Rodamin filter (352-477 nm) we observed mainly epicatechin because the highest fluorescence emission of epicatechin is 320nm, and in GFP filter (457-538nm) we observed quercetin, since its highest fluorescence emission is 540nm. Those were the fluorescence filters that we have available.
5) cell viability test: perform the test at 24, 48 and 72 hrs using the three NPs. This test is essential in order to have an indication of the biocompatibility of the nanocarriers that are expected to be used to deliver the drugs.
Cell viability is an important parameter that is frequently measured in cell culture experiments. It refers to the proportion of cells that are alive and able to carry out their normal functions. It is generally recommended to measure cell viability at 24 hours after treatment. There are several reasons for this.
Firstly, 24 hours is enough time for most cells to recover from any stressors or damage that they may have experienced during the experimental manipulations. This allows the cells to reach a stable state where their viability can be accurately measured.
Secondly, 24 hours is also early enough in the cell culture process to detect any immediate toxic effects of a treatment or compound. This is particularly important if the goal of the experiment is to screen for potential cytotoxicity or cell death-inducing effects. Moreover, 24 hours is a convenient time point for the present experiments because as it is shown in Figure 3, after 175 minutes we have delivered more than 75% of flavonoids in both formulations (NPs and NEQs).
Finally, if we continue the experiment for 48 and 72 hours, we would not be evaluating the effect on cell viability, however, we would evaluate the effect on cell proliferation, which is not the objective of this study.
In conclusion, while there is no strict rule about when to measure cell viability, 24 hours is a commonly used time point that offers a balance between allowing cells to recover from any experimental stressors and detecting immediate toxic effects.
We would like to thank all comments, suggestions, and comments made by the three reviewers, as well as your time and consideration.
We believe that our manuscript would be of interest to the readership of your journal, and we look forward to your favorable consideration. Thank you for your time and attention.

Round 2
Reviewer 1 Report
I consider that the current version is optimal to be published.
Minor editing is required.
Reviewer 2 Report
I agree with publication.
Reviewer 3 Report
Dear Authors,
your manuscript " Novel hybrid Inulin-Soy Protein nanoparticles simultaneously loaded with (-)- Epicatechin and Quercetin and their in-vitro evaluation." can be accept in this form.
Best Regards